# One-Step Diffusion for Detail-Rich and Temporally Consistent Video Super-Resolution

**Yujing Sun**[1,2,*]**, Lingchen Sun**[1,2,*]**, Shuaizheng Liu**[1,2]**,**
**Rongyuan Wu**[1,2]**, Zhengqiang Zhang**[1,2]**, Lei Zhang**[1,2,†]
[1]The Hong Kong Polytechnic University          [2]OPPO Research Institute
{yukki.sun, ling-chen.sun, shuaizheng.liu, rong-yuan.wu, zhengqiang.zhang}@connect.polyu.hk,
cslzhang@comp.polyu.edu.hk
[*]Equal contribution          [†]Corresponding author

## Abstract

It is a challenging problem to reproduce rich spatial details while maintaining temporal consistency in real-world video super-resolution (Real-VSR), especially when we leverage pre-trained generative models such as stable diffusion (SD) for realistic details synthesis. Existing SD-based Real-VSR methods often compromise spatial details for temporal coherence, resulting in suboptimal visual quality. We argue that the key lies in how to effectively extract the degradation-robust temporal consistency priors from the low-quality (LQ) input video and enhance the video details while maintaining the extracted consistency priors. To achieve this, we propose a Dual LoRA Learning (DLoRAL) paradigm to train an effective SD-based one-step diffusion model, achieving realistic frame details and temporal consistency simultaneously. Specifically, we introduce a Cross-Frame Retrieval (CFR) module to aggregate complementary information across frames, and train a Consistency-LoRA (C-LoRA) to learn robust temporal representations from degraded inputs. After consistency learning, we fix the CFR and C-LoRA modules and train a Detail-LoRA (D-LoRA) to enhance spatial details while aligning with the temporal space defined by C-LoRA to keep temporal coherence. The two phases alternate iteratively for optimization, collaboratively delivering consistent and detail-rich outputs. During inference, the two LoRA branches are merged into the SD model, allowing efficient and high-quality video restoration in a single diffusion step. Experiments show that DLoRAL achieves strong performance in both accuracy and speed. Code and models are available at `https://github.com/yjsunnn/DLoRAL`.

## 1 Introduction

Video super-resolution (VSR) aims to reconstruct high-quality (HQ) videos from low-quality (LQ) inputs. Traditional VSR methods typically rely on convolutional neural network (CNN)-based [32, 7] and Transformer-based designs [5, 19], trained with pixel-wise $L_2$ or $L_1$ losses. While effective in some metrics (*e.g.,* PSNR), these methods often produce over-smoothed results without fine details. To improve perceptual quality, generative adversarial network (GAN)-based VSR methods incorporate the adversarial loss [15] during training to encourage sharper details restoration [3, 21, 4, 41]. However, many VSR models [42, 11, 53] are trained under simplified degradation assumptions (*e.g.*, bicubic downsampling), limiting their performance on real-world LQ videos with complex and unknown degradations. Additionally, GAN-based methods can produce unnatural

---

[†]This work is supported by the PolyU-OPPO Joint Innovative Research Center.

39th Conference on Neural Information Processing Systems (NeurIPS 2025).

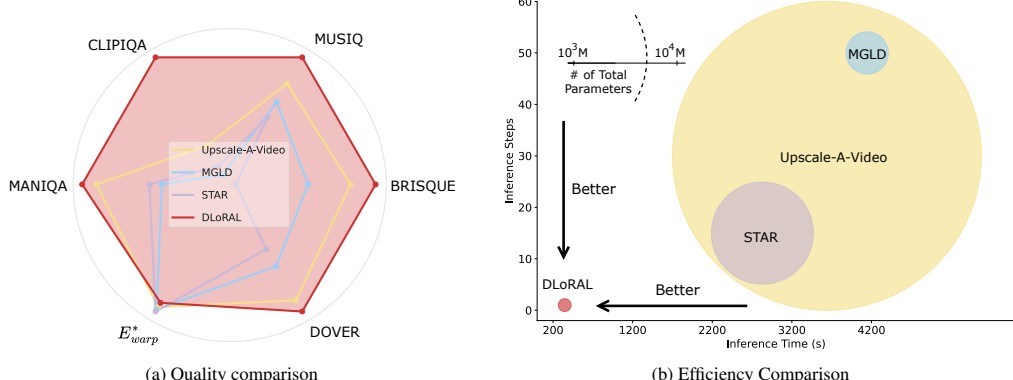

| (a) Quality comparison | (b) Efficiency Comparison |

Figure 1: Quality and efficiency comparison among SD-based Real-VSR methods. (a) Quality comparison on the VideoLQ benchmark [8]. (b) Efficiency comparison tested on an A100 GPU ($512 \times 512$ input with 50 frames for $\times 4$ VSR). DLoRAL achieves the best perceptual quality with only one diffusion step, about $10\times$ faster than Upscale-A-Video [54], MGLD [44], and STAR [40].

artifacts and generalize poorly to diverse video content. Recently, pre-trained diffusion-based text-to-image (T2I) models such as Stable Diffusion (SD) [25, 2] have shown impressive results in real-world image super-resolution (Real-ISR) [31, 38, 48, 26, 37, 27, 34] with realistic textures. One line of research treats the LQ image as a control signal and employs ControlNet-like structures [51] to guide generation [31, 48, 38, 26], and another line of research directly fine-tunes the SD model with LoRA [13] for efficient one-step restoration [37, 27].

The success of SD in Real-ISR inspired exploration of diffusion models for real-world video super-resolution (Real-VSR). Although the powerful generative priors of SD can enhance details, they can introduce inconsistencies among frames when the generated textures sometimes deviate from the content of the LQ inputs [31, 27]. To alleviate this issue, existing SD-based Real-VSR methods typically suppress such fluctuations at the cost of perceptual quality. These methods, such as Upscale-A-Video [54] and MGLD-VSR [44], incorporate temporal modules into pre-trained SD models and adopt frame-wise losses to balance spatial detail and temporal consistency. Despite the significant progress achieved, these methods have two major limitations. First, these approaches optimize detail and consistency jointly in a single model, resulting in suboptimal trade-offs. Improving one objective usually harms the other due to their conflicting nature. Second, the temporal consistency existing in real-world LQ videos is ignored, which can be effectively leveraged to help anchor detail generation on a consistent temporal basis.

To address these issues, we propose a Dual LoRA Learning (DLoRAL) framework for Real-VSR. Our method is built on a one-step residual diffusion model [37, 27], which significantly reduces inference time while maintaining strong generative capability. Inspired by PiSA-SR [27], which learns two LoRA modules to achieve adjustable Real-ISR results, we design two decoupled LoRA branches within the shared diffusion UNet to resolve the conflict between spatial detail and temporal coherence. Specifically, a Consistency-LoRA (C-LoRA) is designed to learn temporal consistency representation, and a Detail-LoRA (D-LoRA) is designed to restore high-frequency spatial details. To exploit the inherent temporal consistency in LQ videos, we introduce a Cross-Frame Retrieval (CFR) module, which extracts structure-aligned temporal features from adjacent degraded frames, helping the model learn degradation-robust representations. CFR not only provides a stable and informative intermediate representation for C-LoRA to build upon, but also serves as the anchor for the subsequent detail enhancement stage to maintain temporal alignment.

Instead of optimizing both objectives jointly, we adopt a dual-stage training strategy. The training begins from the temporal consistency stage, in which we fine-tune C-LoRA and CFR modules using consistency-related losses. In the detail enhancement stage, we freeze C-LoRA and CFR, and train D-LoRA to refine high-frequency details with the additional classifier score distillation (CSD) [27] loss. These two stages are alternatively trained to allow each branch to specialize in its objective. During inference, the two LoRA modules can be integrated in one-step diffusion. As illustrated in Fig. 1, our DLoRAL method achieves both high temporal consistency and superior visual quality,

outperforming previous Real-VSR methods in overall quality, as well as inference speed (about $10\times$ speedup over current methods [54, 44, 40], as illustrated in Fig. 1(b)).

Our main contributions are summarized as follows. (1) We propose a Dual LoRA Learning (DLoRAL) paradigm for Real-VSR, which decouples the learning of temporal consistency and spatial details into two dedicated LoRA modules under a unified one-step diffusion framework. (2) We introduce a Cross-Frame Retrieval (CFR) module to extract degradation-robust temporal priors for Consistency-LoRA (C-LoRA) training, providing structure-aligned intermediate representations that guide the subsequent training of Detail-LoRA (D-LoRA) for high-fidelity restoration. (3) Our DLoRAL model achieves state-of-the-art performance on Real-VSR benchmarks, producing visually realistic frame details and stable temporal consistency.

## 2 Related Work

**Real-World VSR.** Conventional VSR methods [32, 11, 16] typically rely on simply synthesized data (*e.g.*, bicubic downsampling), leading to a significant performance gap when applied to real-world videos. Early works [45, 35] addressed this by collecting real-world LQ-HQ video pairs, such as the iPhone-captured dataset [45]. However, these datasets are limited by device bias and scalability. The following works [33, 8] simulated realistic degradations by combining blur, noise, and compression, while others enhanced robustness through architectural design. For instance, RealVSR [45] introduces a domain adaptation mechanism that aligns feature distributions between synthetic and real domains through adversarial learning. RealBasicVSR [8] proposes a degradation modeling framework that refines the restoration process through iterative correction modules. Despite these advances, existing methods still struggle to recover fine details and generalize across diverse real-world scenarios, often producing over-smoothed outputs.

**Diffusion Based Real-VSR**. Recent advances in diffusion models for image restoration [1, 10, 23, 49, 50] have inspired the extension to Real-VSR tasks [54, 44, 17, 40]. A common approach is to adapt pre-trained T2I models by injecting temporal modules to ensure both perceptual quality and temporal consistency. For example, Upscale-A-Video [54] integrates temporal layers into the pre-trained diffusion model and proposes a flow-guided recurrent latent propagation module. MGLD-VSR [44] guides the diffusion process with a motion-guided loss and inserts a temporal module into the diffusion decoder. The other directions include decomposing the complex learning burden into staged training phases [17] and reformulating attention mechanisms in diffusion transformers [30] to process videos of arbitrary length. Rather than leveraging the pre-trained T2I model, STAR [40] leverages compressed temporal representations from text-to-video (T2V) models [2].Despite these efforts, balancing spatial detail and temporal consistency remains a key challenge. Most existing methods enforce frame-level constraints to improve consistency by sacrificing visual fidelity. In this work, we propose a decoupled learning strategy: first learning degradation-robust temporal priors from LQ inputs then guiding HQ generation with these features. This design ensures both high-quality detail restoration and stable temporal coherence.

**Real-VSR Paradigms.** Recent VSR methods follow two main paradigms: sliding-window-based [32, 11, 42, 16] and recurrent-based [6, 7, 5, 20, 54, 44, 40]. Sliding-window-based methods reconstruct each output frame using a set of neighboring frames, capturing fine-grained local details and short-term temporal dependencies. In contrast, recurrent-based methods propagate features across frames sequentially, offering higher efficiency, but are prone to error accumulation and detail degradation. Most diffusion-based Real-VSR methods [54, 44, 40] adopt the recurrent design for its inference efficiency. In this work, we build on a sliding-window framework to better preserve spatial and temporal details. To mitigate the computational overhead, we adopt a one-step diffusion strategy that eliminates redundancy while maintaining high reconstruction quality.

## 3 Methodology

### 3.1 Preliminary

Diffusion models [25] simulate a forward process where a clean latent code $z_0$ is gradually noised into $z_t$ using Gaussian noise: $z_t = \sqrt{\bar{\alpha}_t} \cdot z_0 + \sqrt{1 - \bar{\alpha}_t} \cdot \epsilon$, with $\epsilon \sim \mathcal{N}(0, I)$ and $\bar{\alpha}_t$ following a predefined schedule. During training, a model $\epsilon_\theta(t, z_t)$ is trained to predict the added noise at each timestep $t$. During inference, $z_0$ is recovered from pure noise $z_T \sim \mathcal{N}(0, I)$ by iterative denoising. However,

this multi-step process is slow and stochastic, limiting its efficiency and stability in super-resolution (SR) tasks that demand fast and reliable reconstruction. To address this issue, recent works [37, 27] propose one-step diffusion that skips iterative sampling by directly refining an LQ latent into its HQ counterpart. To further improve controllability, PiSA-SR [27] introduces a residual learning formulation that allows the model to focuses on high-frequency corrections:

$$z^{HQ} = z^{LQ} - \epsilon_\theta(z^{LQ}) \tag{1}$$

where $z^{LQ}$ and $z^{HQ}$ represent the latent codes of LQ and HQ respectively.

Most existing VSR methods [54, 44, 40] rely on multi-step diffusion, resulting in high computational cost. In this work, we make the first attempt to apply a one-step diffusion framework to VSR, improving efficiency while preserving restoration quality by adapting the residual learning formulation in Eq. (1) to accelerate convergence. To this end, we introduce VSR-specific modules and learning strategies to produce detail-rich and temporally consistent results.

### 3.2 Dual LoRA Learning Network for Real-VSR

**Motivation.** There is a fundamental challenge in Real-VSR: how to balance the preservation of spatial details and the enforcement of temporal consistency. To simultaneously achieve both objectives, we begin by analyzing the characteristics of real-world LQ videos and the limitations of current SD-based VSR methods, which motivate the design of our proposed framework.

- *Temporal Consistency in Degraded Videos.* Despite degradations, such as noise, blur, and compression, real-world LQ videos retain stable information across frames, preserving inherent structural and semantic consistency. Leveraging these consistent representations provides a strong foundation for reconstructing HQ videos with realistic details and temporal coherence. To exploit this, we propose a Cross-Frame Retrieval (CFR) module to aggregate complementary information across frames to enhance consistency. In addition, we design a Consistency-LoRA (C-LoRA) to further improve reconstruction by reinforcing temporal alignment and structural integrity. This stage lays the groundwork for more accurate guidance in the subsequent detail enhancement phase.
- *Conflict in Optimizing Details and Consistency.* To adapt pre-trained diffusion models for VSR and balance spatial detail and temporal coherence, existing methods [25, 46] typically introduce trainable layers optimized jointly with diffusion and temporal losses. However, detail generation and consistency preservation are inherently conflicting objectives, and joint optimization often results in suboptimal trade-offs. To address this, we propose two decoupled weight spaces: one for temporal consistency modeling and another for detail enhancement. Rather than training two networks, which is costly, we adopt a decoupled scheme inspired by PiSA-SR [27], embedding two specialized LoRA branches into a shared SD UNet. This lightweight design enables alternative refinement, allowing each branch to focus on its objective.

**Framework Overview.** Building on the above insights, we design a Dual LoRA Learning (DLoRAL) framework to generate HQ video outputs from degraded inputs. Given an LQ sequence of $N$ frames, $\mathbf{I}^{LQ} = \{I_n^{LQ} \mid n = 1, \ldots, N\}$, our model $G_\theta$ generates a corresponding HQ sequence $\mathbf{I}^{HQ} = \{I_n^{HQ} \mid n = 1, \ldots, N\}$. To utilize information from neighboring frames, we adopt a sliding-window strategy [32, 11, 16], where each HQ frame $I_n^{HQ}$ is generated from two adjacent LQ frames: the current $n$-th frame $I_n^{LQ}$ and its preceding frame $I_{n-1}^{LQ}$. For the first frame $I_1^{LQ}$, which lacks a previous frame, we adopt a self-replication approach to generate $I_1^{LQ}$.

As illustrated in Fig. 2, our generator $G_\theta$ leverages the pre-trained SD model, which consists of a VAE encoder $E_\theta$, an SD UNet $\epsilon_\theta$, and a VAE decoder $D_\theta$. Our DLoRAL framework employs two specialized training stages, *i.e.,* temporal consistency stage and detail enhancement stage. In the temporal consistency stage, the CFR module retrieves inter-frame relevant information from degraded inputs, then the UNet is finetuned by C-LoRA for further reinforcement of temporal alignment. In the detail enhancement stage, D-LoRA is optimized to improve spatial visual quality. These two stages are trained alternately in an iterative manner to progressively refine both temporal consistency and spatial quality, ultimately leading to coherent and detail-preserved video restoration. During inference, the C-LoRA and D-LoRA are merged into the SD UNet to ensure efficient deployment.

**Temporal Consistency Stage.** This stage is to establish a temporally coherent and robust representation from the LQ video sequence $\mathbf{I}^{LQ}$ before enhancing details. This stage involves two main steps: temporal feature fusion using a CFR module and fine-tuning the SD UNet to improve consistency.

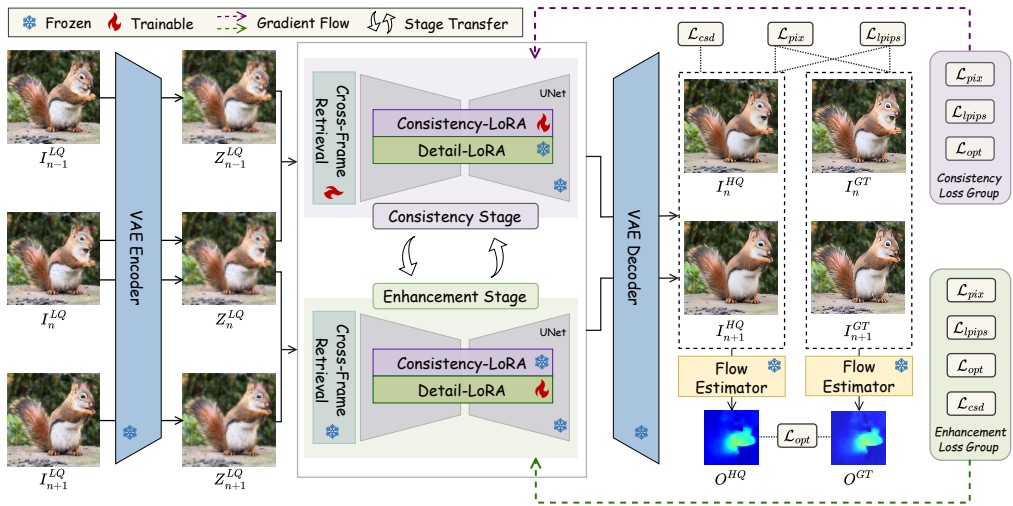

Figure 2: The training pipeline of our proposed DLoRAL. The Cross-Frame Retrieval (CFR) and Consistency-LoRA (C-LoRA) modules are optimized in the consistency stage, while the Detail-LoRA (D-LoRA) is optimized in the enhancement stage. Both stages are alternately trained to ensure temporal coherence and visual quality.

To unlock the inherent consistency among degraded inputs, CFR improves the current latent representation $z_n^{LQ}$ by employing a specialized attention mechanism that integrates complementary information from the previous latent feature $z_{n-1}^{LQ}$. Specifically, with encoded features $z_n^{LQ}$ and $z_{n-1}^{LQ}$, CFR first warps them into the same coordinate space with SpyNet [24] following a common frame alignment procedure [32, 44] (denoted as $F_{wp}$). The current latent features $z_n^{LQ}$ and aligned latent features $F_{wp}(z_{n-1}^{LQ})$ are then projected into query ($Q_n$), key ($K_{n-1}$), and value ($V_{n-1}$) embeddings through $1 \times 1$ convolutions (denoted as $\circ$) parameterized by $W_Q$, $W_K$, and $W_V$, as shown below:

$$Q_n = W_Q \circ z_n^{LQ}, \quad K_{n-1} = W_K \circ F_{wp}(z_{n-1}^{LQ}), \quad V_{n-1} = W_V \circ F_{wp}(z_{n-1}^{LQ}). \tag{2}$$

With obtained embeddings extracted from adjacent frames, CFR employs two mechanisms to enhance fusion quality. First, for each query position $p$, it selectively attends to only the top-k most similar positions (denoted as $F_{topk}[p]$) in the aligned previous frame, avoiding perturbations from uncorrelated noises. Second, for each query position $p$, a learnable threshold $\tau_n[p]$ is predicted via a lightweight MLP. It dynamically adapts to regional characteristics - enforcing stricter filtering in detail-rich areas while being more permissive in flat regions, ensuring that only confident matches could contribute to the final fusion. The fused feature $\bar{z}_n^{LQ}[p]$ is computed as:

$$\bar{z}_n^{LQ}[p] = z_n^{LQ}[p] + \sum_{q \in F_{topk}[p]} \phi\left(\frac{\langle Q_n[p], K_{n-1}[q]\rangle}{\sqrt{d}} - \tau_n[p]\right) \cdot V_{n-1}[q], \tag{3}$$

where $\phi(\cdot)$ is a non-negative gating function (*e.g.*, ReLU [12]), and $d$ is the channel dimension.

The latent feature $\bar{z}_n^{LQ}$ is then processed by the UNet to generate the HQ latent $z_n^{HQ}$. In this stage, only the C-LoRA is trainable, while D-LoRA remains frozen. The final HQ frame is reconstructed via the VAE decoder by $I_n^{HQ} = D_\theta(z_n^{HQ})$. All trainable components in this stage, including the CFR module and C-LoRA, are optimized using the consistency loss $\mathcal{L}_{\text{cons}}$, which is designed to ensure both the quality of individual frames and the temporal consistency across the sequence. It combines the pixel-level loss ($\mathcal{L}_{\text{pix}}$), LPIPS loss ($\mathcal{L}_{\text{lpips}}$), and optical flow loss ($\mathcal{L}_{\text{opt}}$), as shown below:

$$\begin{aligned}
\mathcal{L}_{\text{cons}} &= \lambda_{\text{pix}}\mathcal{L}_{\text{pix}} + \lambda_{\text{lpips}}\mathcal{L}_{\text{lpips}} + \lambda_{\text{opt}}\mathcal{L}_{\text{opt}}, \\
\mathcal{L}_{\text{opt}} &= \left\|O_n^{HQ} - O_n^{\text{GT}}\right\|_1 = \left\|F(I_n^{HQ}, I_{n+1}^{HQ}) - F(I_n^{\text{GT}}, I_{n+1}^{\text{GT}})\right\|_1.
\end{aligned} \tag{4}$$

Here, the $\ell_2$ loss is adopted as the $\mathcal{L}_{\text{pix}}$, and $\mathcal{L}_{\text{opt}}$ measures the $L_1$ distance between optical flow maps estimated from generated and ground-truth frame pairs, promoting motion alignment and smooth

transitions. The loss weights $\lambda_{\text{pix}}$, $\lambda_{\text{lpips}}$, and $\lambda_{\text{opt}}$ are empirically set to balance spatial accuracy, perceptual quality, and temporal consistency.

**Detail Enhancement Stage.** Different from the temporal consistency stage, which yields aligned and coherent latent representations, the detail enhancement stage focuses on restoring high-frequency visual details. In this stage, adjacent latent features $z_{n-1}^{LQ}$ and $z_n^{LQ}$ are processed by the frozen CFR module to reapply the learned alignment and fusion, thus the temporal consistency learned in the consistency stage is maintained without introducing new variations.

The resulting temporally enriched latent $\bar{z}_n^{LQ}$ is then fed into the diffusion UNet $\epsilon_\theta$. We employ a decoupled finetuning strategy: only the D-LoRA parameters, responsible for detail synthesis, are trainable, while the C-LoRA parameters, associated with consistency, remain frozen. This setting allows the D-LoRA to focus solely on detail synthesis without compromising the temporal structure previously established. The output HQ latent $z_n^{HQ}$ is then decoded using the frozen decoder $D_\theta$ to produce the final super-resolved frame $I_n^{HQ}$.

To guide this detail enhancement while preserving the structure learned previously, the loss function $\mathcal{L}_{\text{enh}}$ combines several components as follows:

$$\mathcal{L}_{\text{enh}} = \lambda_{\text{pix}}\mathcal{L}_{\text{pix}} + \lambda_{\text{lpips}}\mathcal{L}_{\text{lpips}} + \lambda_{\text{opt}}\mathcal{L}_{\text{opt}} + \lambda_{\text{csd}}\mathcal{L}_{\text{csd}}. \tag{5}$$

We retain $\mathcal{L}_{\text{pix}}$, $\mathcal{L}_{\text{lpips}}$, and $\mathcal{L}_{\text{opt}}$ used in the consistency stage (as Eq. (4)), serving as anchors to maintain spatial fidelity and motion coherence. Furthermore, we introduce the Classifier Score Distillation (CSD) loss [27], $\mathcal{L}_{\text{csd}}$, which encourages the generation of richer and finer details.

## 3.3   Training and Inference

**Dynamic Dual-Stage Training.** We adopt a dynamic dual-stage training scheme. The training begins with the consistency stage, aiming at learning degradation-robust features and establishing strong temporal coherence among frames. In this stage, only the CFR and C-LoRA modules are trainable, while the D-LoRA is fixed. Once the model converges in the consistency stage, the training switches to refine high-frequency spatial details, guided by $\mathcal{L}_{\text{enh}}$ with the additional CSD loss. In this stage, only the D-LoRA parameters are trainable, while the CFR module and C-LoRA are fixed. Such an alternative training is iterated, allowing the model to dynamically converge toward a solution that balances temporal coherence and visual fidelity.

**Smooth Transition Between Training Stages.** Compared to the consistency stage, the enhancement stage introduces an additional loss function $\mathcal{L}_{\text{csd}}$ for enriching semantic details. Directly switching between the full loss functions $\mathcal{L}_{\text{cons}}$ and $\mathcal{L}_{\text{enh}}$ can lead to instability due to the abrupt change in learning targets. To prevent this, we employ a re-weighting strategy that progressively shifts the loss objective, ensuring a smooth transition between stages. Taking the transition from the consistency stage to the enhancement stage as an example, after the consistency stage, the two loss functions are interpolated as the optimization objective for a warm-up phase of $s_t$ steps, as shown below:

$$\mathcal{L}(s) = (1 - \frac{s}{s_t}) \cdot \mathcal{L}_{\text{cons}} + \frac{s}{s_t} \cdot \mathcal{L}_{\text{enh}}, \quad s \in [0, s_t], \tag{6}$$

where $s$ denotes the current step within the transition. Symmetric interpolation is applied when we switch back from the enhancement stage to the consistency stage.

**Inference Phase.** At test time, both C-LoRA and D-LoRA are activated and merged into the frozen diffusion UNet. A single diffusion step is used to enhance the LQ input to HQ video frames.

## 4   Experiment

### 4.1   Experimental Settings

**Implementation Details.** We adopt the pre-trained Stable Diffusion V2.1 as the backbone of denoising U-Net. Training is carried out with a batch size of 16, a sequence length of 3, and a video resolution of $512 \times 512$. All models are trained using the PyTorch framework on 4 NVIDIA A100 GPUs. We use Adam optimizer with an initial learning rate of $5 \times 10^{-5}$. For inference, both C-LoRA and D-LoRA are activated simultaneously in a frozen UNet. Videos are processed in sliding sequences to fit GPU memory limits.

**Training Datasets.** To support the decoupled training design of our DLoRAL framework, we construct two training datasets for the consistency and enhancement stages, respectively.

For the *consistency stage*, the training data needs to contain realistic motion while maintaining reasonable image quality. To this end, we select 44,162 high-quality frames from the REDS dataset [22], which offers professionally captured sequences with rich dynamics, and a curated set of videos [39] from Pexels[1], chosen based on aesthetic and temporal smoothness criteria. These sequences provide necessary temporal priors for learning degradation-robust representations.

For the *enhancement stage*, the training data should prioritize visual quality. Thus, we select the LSDIR [18] dataset, known for its rich textures and more fine-grained details than existing public video datasets. To preserve the learned consistency modeling capability and enable the optical flow regularization among frames, we generate simulated video sequences based on LSDIR. Specifically, for each ground-truth image in LSDIR, we apply random pixel-level translations to it to generate multiple shifted images. The resulting pseudo-video sequences inherently support consistency constraints through synthetic motion, while surpassing real video datasets in visual quality.

The data in both stages are degraded using the RealESRGAN [33] degradation pipeline. We apply identical degradation parameters across frames within the same video, while using random parameters for different video sequences.

**Testing Datasets.** We evaluate our method on both synthetic and real-world datasets, including UDM10 [47], SPMCS [28], RealVSR [45], and VideoLQ [8]. Among them, UDM10 contains 10 sequences, each having 32 frames. SPMCS contains 30 sequences, each having 31 frames. RealVSR contains 50 real-world sequences, each having 50 frames. VideoLQ contains 50 real-world sequences with complex degradations. For the synthetic dataset (UDM10 and SPMCS), we synthesize LQ-HQ pairs following the same degradation pipeline in training. For real-world datasets (RealVSR and VideoLQ), we directly adopt the given LQ-HQ pairs.

**Evaluation Metrics.** A set of full-reference and no-reference metrics are selected to evaluate different real-world VSR methods. The full-reference metrics include PSNR and SSIM, and perceptual quality with LPIPS [52] and DISTS [9]. No-reference quality assessment involves MUSIQ [14], MANIQA [43], CLIPIQA [29], and the video quality assessment metric DOVER [36]. Compared to Real-ISR, Real-VSR places greater emphasis on temporal consistency. Following prior works [44, 53], we use the average warping error $E_{warp}^*$ to quantitatively assess temporal consistency: $E_{warp}^* = \frac{1}{N-1} \sum_{i=1}^{N-1} ||I_{i+1}^{HQ} - F_{wp}(I_i^{HQ})||_1$. For the test datasets with GT, optical flow in $F_{wp}$ is estimated from GT frames. For real-world datasets without GT (*e.g.*, VideoLQ test set), we use the flow estimated from predicted frames.

## 4.2 Experimental Results

To demonstrate the effectiveness of our DLoRAL algorithm, we compare it with seven represnetative and state-of-the-art methods, including three Real-ISR models (RealESRGAN [33], StableSR [31], and the one-step model OSEDiff [37]), a discriminative VSR model (RealBasicVSR [8]), and three diffusion-based VSR models (Upscale-A-Video [54], MGLD-VSR [44] and STAR [40]).

**Quantitative Comparison.** We show the quantitative comparison on both synthetic and real-world video benchmarks (where real-world testing videos were centrally cropped to $128 \times 128$ resolution) in Tab 1, from which several key observations can be made. First, non-diffusion-based methods (*e.g.*, RealESRGAN and RealBasicVSR) perform worse than diffusion-based methods on no-reference perceptual quality metrics, such as MUSIQ and CLIPIQA, mainly because they lack the strong image priors provided by pre-trained SD models, leading to over-smoothed results. Second, SD-based Real-ISR methods (StableSR and OSEDiff) can achieve comparable or even better perceptual quality scores than existing Real-VSR methods. In particular, OSEDiff achieves the best DOVER scores on both the UDM10 and SPMCS datasets. However, its warping error evaluated by $E_{warp}^*$ is worse. This is because the Real-ISR methods generate details for each frame without considering the inter-frame consistency. Finally, compared to the existing Real-VSR methods, our DLoRAL consistently ranks first or second across a range of perceptual quality metrics, including LPIPS, DISTS, MUSIQ, CLIPIQA, MANIQA, and DOVER, demonstrating its strong alignment with human perception. At the same time, DLoRAL does not compromise temporal consistency, as evidenced

---

[1] https://www.pexels.com/

| Datasets | Metrics | Real-ISR Methods | | | Real-VSR Methods | | | | |
|---|---|---|---|---|---|---|---|---|---|
| | | RealESRGSN | StableSR | OSEDiff | RealBasicVSR | Upscale-A-Video | MGLD | STAR | **DLoRAL** |
| UDM10 | PSNR ↑ | 21.345 | 22.042 | 23.761 | **24.334** | 22.364 | 24.192 | **24.451** | 23.975 |
| | SSIM ↑ | 0.565 | 0.568 | 0.696 | **0.723** | 0.584 | 0.685 | **0.714** | 0.710 |
| | LPIPS ↓ | 0.451 | 0.455 | 0.367 | 0.363 | 0.410 | **0.335** | 0.417 | **0.327** |
| | DISTS ↓ | **0.175** | 0.185 | **0.175** | 0.204 | 0.198 | 0.176 | 0.230 | 0.179 |
| | BRISQUE ↓ | 29.843 | 26.310 | 20.718 | **14.129** | 17.607 | 22.701 | 36.910 | **16.250** |
| | MUSIQ ↑ | 49.838 | 47.805 | **63.146** | 62.360 | 61.046 | 61.309 | 40.789 | **65.620** |
| | CLIPIQA ↑ | 0.474 | 0.445 | **0.574** | 0.474 | 0.445 | 0.453 | 0.267 | **0.652** |
| | MANIQA ↑ | 0.330 | 0.319 | **0.334** | 0.330 | 0.318 | 0.291 | 0.244 | **0.373** |
| | $E^*_{\text{warp}}$ ↓ | 7.580 | 8.440 | 5.220 | 4.670 | 5.790 | **4.610** | **3.510** | 4.720 |
| | DOVER ↑ | 36.860 | 30.470 | **48.404** | 37.572 | 37.694 | 40.045 | 30.384 | **42.871** |
| SPMCS | PSNR ↑ | **21.660** | 19.260 | 20.650 | **21.580** | 19.030 | 21.260 | 20.730 | 21.240 |
| | SSIM ↑ | 0.569 | **0.585** | **0.696** | 0.545 | 0.386 | 0.515 | 0.489 | 0.524 |
| | LPIPS ↓ | 0.444 | 0.432 | **0.354** | 0.404 | 0.485 | 0.384 | 0.606 | **0.375** |
| | DISTS ↓ | 0.246 | 0.235 | **0.229** | 0.237 | 0.274 | 0.234 | 0.342 | **0.222** |
| | BRISQUE ↓ | 25.240 | 26.310 | 19.471 | 12.048 | **19.784** | 23.184 | 27.902 | **11.030** |
| | MUSIQ ↑ | 53.221 | 47.805 | 64.619 | 66.683 | **66.912** | 65.079 | 33.247 | **67.390** |
| | CLIPIQA ↑ | 0.515 | 0.445 | **0.526** | 0.515 | 0.517 | 0.437 | 0.240 | **0.581** |
| | MANIQA ↑ | 0.308 | 0.319 | 0.308 | 0.308 | **0.443** | 0.312 | 0.237 | **0.340** |
| | $E^*_{\text{warp}}$ ↓ | 7.570 | 8.430 | 7.500 | 5.400 | 7.570 | **4.410** | **4.080** | 6.250 |
| | DOVER ↑ | 32.151 | 30.470 | **40.160** | 30.953 | 32.151 | 31.118 | 17.220 | **34.895** |
| RealVSR | PSNR ↑ | **21.340** | 18.950 | 19.920 | **22.270** | 20.060 | 21.120 | 15.080 | 20.360 |
| | SSIM ↑ | 0.565 | 0.583 | 0.588 | **0.720** | 0.591 | **0.646** | 0.433 | 0.606 |
| | LPIPS ↓ | 0.451 | 0.225 | 0.282 | **0.193** | 0.263 | **0.219** | 0.409 | 0.242 |
| | DISTS ↓ | 0.175 | 0.154 | 0.164 | 0.160 | 0.158 | **0.151** | 0.279 | **0.150** |
| | BRISQUE ↓ | 29.843 | 28.250 | 31.794 | 30.362 | **25.476** | 39.082 | 62.750 | **27.893** |
| | MUSIQ ↑ | 49.838 | 69.962 | 64.101 | **71.413** | 67.714 | 70.734 | 67.947 | **70.908** |
| | CLIPIQA ↑ | 0.474 | **0.612** | 0.546 | 0.370 | 0.436 | 0.530 | 0.532 | **0.617** |
| | MANIQA ↑ | 0.330 | 0.345 | 0.341 | 0.384 | **0.414** | **0.496** | 0.438 | 0.386 |
| | $E^*_{\text{warp}}$ ↓ | 17.580 | 25.010 | 18.300 | 18.720 | **18.200** | 19.210 | 24.600 | **17.300** |
| | DOVER ↑ | 36.860 | **46.846** | 42.138 | 46.439 | 36.136 | 42.044 | 30.214 | **49.646** |
| VideoLQ | BRISQUE ↓ | 29.605 | **22.337** | 26.403 | 24.790 | 25.101 | 29.606 | 42.582 | **23.039** |
| | MUSIQ ↑ | 53.138 | 52.975 | 58.959 | **59.475** | 57.489 | 53.092 | 49.305 | **63.846** |
| | CLIPIQA ↑ | 0.334 | 0.478 | **0.499** | 0.393 | 0.377 | 0.315 | 0.333 | **0.567** |
| | MANIQA ↑ | 0.232 | 0.278 | 0.254 | 0.312 | **0.328** | 0.254 | 0.268 | **0.344** |
| | $E^*_{\text{warp}}$ ↓ | 7.580 | 8.430 | 8.406 | 8.108 | 7.586 | **7.409** | **7.280** | 7.897 |
| | DOVER ↑ | 28.400 | 30.470 | **37.580** | 34.772 | 36.860 | 31.899 | 29.400 | **38.505** |

Table 1: Comparison of various Real-ISR and Real-VSR methods across different datasets. The best and second best results of each metric are highlighted in **red** and **blue**, respectively.

by its superior performance on the $E^*_{warp}$ metric. For example, on the RealVSR dataset, DLoRAL achieves state-of-the-art results in DISTS, CLIPIQA, and DOVER, while ranking among the top in $E^*_{warp}$, highlighting its ability to produce visually pleasing and temporally coherent outputs.

It should be mentioned that although $E^*_{warp}$ is widely used to assess temporal consistency, it does not correlate well with human perception. For example, blurry Real-VSR outputs can achieve lower warping errors but exhibit poorer visual quality. DLoRAL may report slightly larger $E^*_{warp}$ values than some methods (*e.g.*, STAR), but this is because DLoRAL better preserves fine details that the warping error metric tends to penalize.

**Qualitative Comparison.** To further demonstrate the effectiveness of DLoRAL, we visualize the Real-VSR results in Fig. 3. One can see that DLoRAL can remove complex spatial-variant degradations and generate realistic details, significantly outperforming other Real-VSR models. Specifically, for the severely degraded facial region (first row), RealBasicVSR fails to reconstruct the facial structural, and Upscale-A-Video and STAR lose facial details. MGLD produces sharper outputs, but suffers from severe structural distortions, particularly around the eye regions. In contrast, DLoRAL successfully recovers fine facial features while maintaining structural integrity. The second row highlights the performance in texture reconstruction, where our method restores sharper and more legible texture patterns compared to the blurry or distorted outputs from other algorithms.

To better compare the consistency, we plot the temporal profiles of the VSR results produced by competing methods Fig. 4. Real-ISR approaches such as StableSR and OSEDiff restore sharper details but suffer from severe temporal instability, as shown by the erratic fluctuations in their profiles,

resulting in unpleasant flickering that harms the video quality. On the other hand, while existing Real-VSR methods can offer better temporal consistency than Real-ISR methods, this comes at the cost of blurred details (see the results of Upscale-A-Video, MGLD and STAR in the left case of Fig. 4) or intra-frame artifacts (see the results of RealBasicVSR, Upscale-A-Video and STAR in the right case of Fig. 4). In comparison, our DLoRAL produces smooth and stable transitions across frames, as reflected by its consistent temporal profiles. This qualitative evidence aligns with our quantitative results, demonstrating DLoRAL's ability to preserve fine visual details while ensuring natural temporal consistency. More visual comparisons can be found in the **Appendix**.

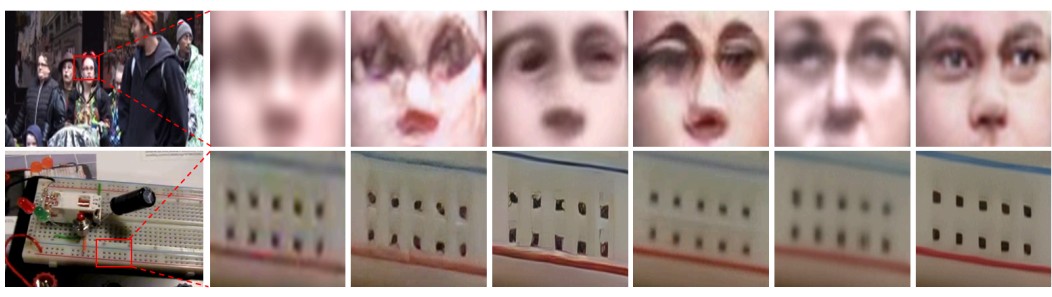

| LQ | Zoomed LQ | RealBasicVSR | Upscale-A-Video | MGLD | STAR | DLoRAL |

Figure 3: Qualitative comparison of VSR models on real-world VideoLQ dataset.

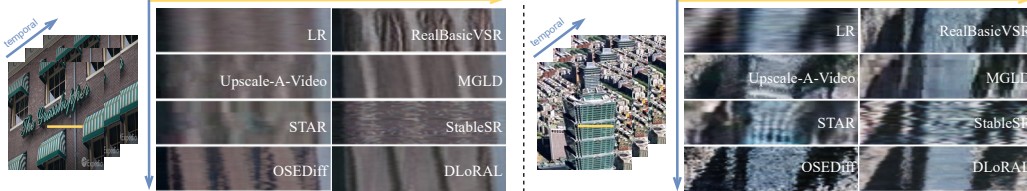

Figure 4: Temporal profiles of competing Real-ISR and Real-VSR methods.

| | Real-ISR Methods | | Real-VSR Methods | | | |
|---|---|---|---|---|---|---|
| | StableSR | OSEDiff | Upscale-A-Video | MGLD | STAR | **DLoRAL** |
| Inference Step | 200 | 1 | 30 | 50 | 15 | 1 |
| Inference Time (s/50 frames) | 32800 | 340 | 3640 | 4146 | 2830 | 346 |
| # Total Param (M) | 1150 | 1294 | 14442 | 1430 | 2492 | 1300 |

Table 2: Complexity comparison among different methods. All methods are evaluated using 50 $512 \times 512$ frames for the $\times 4$ VSR task. Inference time is measured on an A100 GPU and includes the entire pipeline: data loading, processing, and result storage.

**Complexity Comparison.** We compare the inference steps, model size, and inference time of competing diffusion-based models in Tab. 2. The inference time of the whole pipeline (including data loading, data processing, and result storage) is reported, which is measured on the $\times 4$ VSR task with 50 frames of $512 \times 512$ LQ images on a single NVIDIA A100 80G GPU. Compared with Real-ISR methods, DLoRAL (346s) achieves a strong balance between quality and complexity, delivering superior visual quality and temporal consistency while maintaining a similar speed to OSEDiff (340s). Among the Real-VSR methods, DLoRAL achieves the fastest inference time and the lowest parameter count, benefiting from its efficient one-step design. Specifically, DLoRAL is more **10× faster** than Upscale-A-Video, MGLD, and **8× faster** than STAR, while maintaining superior visual quality.

**Ablation Study.** To validate the effectiveness of the proposed components in our model, we conduct ablation studies by selectively removing each of the three key modules: (i) CFR, (ii) C-LoRA, and (iii) D-LoRA, while keeping all other settings identical. For this analysis, we adopt VideoLQ4[2], a subset of four representative sequences with diverse scenes and motions from the VideoLQ dataset. As summarized in Tab. 3, removing either CFR or C-LoRA leads to weaker temporal consistency (*i.e.*, higher warping error), indicating their complementary roles in maintaining temporal coherence. In contrast, removing D-LoRA significantly impairs all perceptual metrics, confirming its core contribution to fine-grained detail enhancement. Further ablations are provided in the **Appendix**.

---

[2]Specifically, VideoLQ4 contains the 013, 015, 020, and 041 clips, each consisting of 100 frames.

| | MUSIQ ↑ | CLIP-IQA ↑ | MANIQA ↑ | $E_{\text{warp}}^* \downarrow$ |
|---|---|---|---|---|
| Ours (Full) | **66.6174** | 0.5475 | **0.3791** | $1.51 \times 10^{-3}$ |
| W/o CFR | 64.5732 | 0.5148 | 0.3386 | $1.58 \times 10^{-3}$ |
| W/o C-LoRA | 64.2623 | **0.5492** | 0.3520 | $1.61 \times 10^{-3}$ |
| W/o D-LoRA | 54.0769 | 0.3654 | 0.2471 | $\mathbf{1.48 \times 10^{-3}}$ |

Table 3: Ablation study on key modules on VideoLQ4 dataset.

**User Study.** We also conduct a user study to further examine the effectiveness of DLoRAL in comparison with existing RealVSR methods. We invited ten volunteers to participate in a user study. Our DLoRAL method was compared with the other three diffusion-based Real-VSR methods: Upscale-A-Video [54], MGLD [44] and STAR [40]. We randomly selected 12 real-world LQ videos with complex degradations and motions from the VideoLQ dataset [8], whose scenes are shown in Fig. 5(a). Each LQ video and its corresponding HQ videos generated by the competing Real-VSR methods were presented to the participants who were asked to select the best HQ result by considering two equally weighted factors: the perceptual quality and temporal consistency of the video.

The results of the user study are shown in Fig. 5(b). DLoRAL received **93** votes, significantly outperforming the other methods, with MGLD, STAR, and Upscale-A-Video receiving 14, 8, and 5 votes, respectively. This overwhelming preference for DLoRAL highlights its effectiveness in addressing the challenges of real-world video restoration. Note that the selected videos include a variety of motion scenarios. In scenarios with complex motion, DLoRAL is able to achieve superior visual quality while maintaining temporal consistency comparable to other methods. In relatively static scenes, DLoRAL demonstrates stable temporal consistency along with equally sharp and clear visual quality.

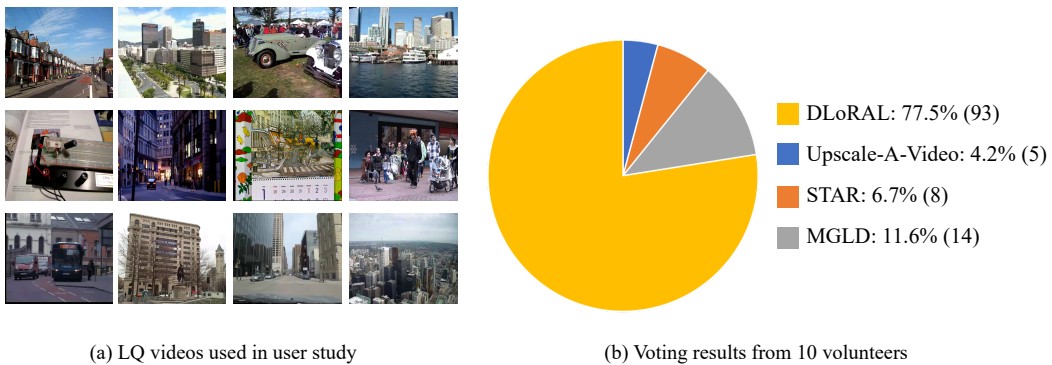

(a) LQ videos used in user study      (b) Voting results from 10 volunteers

Figure 5: LQ videos used in our user study and the voting results.

## 5 Conclusion

We proposed DLoRAL to achieve temporally consistent and detail-rich Real-VSR results. To effectively extract degradation-robust temporal priors from low-quality input videos while enhancing details without compromising these priors, we first developed a CFR module and a consistency-LoRA to generate robust temporal representations, and then developed a detail-LoRA to enhance spatial details. We optimized these two objectives alternatively and iteratively, where the results of the previous stage served as an anchor to provide priors for the next stage. The resulting DLoRAL model demonstrated significantly superior performance to previous Real-VSR methods, achieving rich spatial details without compromising the temporal coherence.

**Limitations**. Despite its strong performance, DLoRAL still has certain limitations. First, since it inherits the $8\times$ downsampling VAE from SD, DLoRAL faces difficulties in restoring very fine-scale details such as small texts. Second, this heavy compression of VAE may disrupt temporal coherence, making it harder to extract robust consistency priors. A VAE specifically designed for Real-VSR tasks could help to address these issues. We leave this challenge for future investigation.

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
