# OpenReview forum: "One-Step Diffusion for Detail-Rich and Temporally Consistent Video Super-Resolution"
_NeurIPS.cc/2025/Conference — NeurIPS 2025 poster_

### Official Review · Reviewer_1Rdb · 2025-06-30

**Clarity:** 3
**Significance:** 3
**Originality:** 2
**Rating:** 5
**Confidence:** 4

**Summary:**

This paper proposes a new diffusion-based video super-resolution pipeline that synthesizes high-resolution videos in one-step diffusion sampling. The author proposes a dual-LoRA learning scheme, consisting of one for retrieving information across frames and one for enhancing video consistency. Extensive results show the effectiveness of their pipeline.

**Questions:**

I wonder why the author selects image diffusion model as their solution instead of video diffusion models, which are effective in solving video tasks.

**Ethical Concerns:**

["NO or VERY MINOR ethics concerns only"]

**Final Justification:**

I want to raise my score to accept as the author has addressed my concerns and questions through the rebuttal period.

**Limitations:**

yes

**Quality:**

3

**Strengths And Weaknesses:**

This paper is well-organized and easy to follow. The results in the video demo are convincing and impressive. However, at the same time, I have some concerns about it:
1. My main concern about this paper is the novelty. The core contribution of this work is the dual-stage LoRA tuning. However, the loss functions of the proposed dual-stage training are very similar, except that the consistency stage utilizes an additional classifier score distillation loss. However, this loss term is not well introduced in the manuscript, and I cannot easily find how and why such a design brings performance gain against existing methods.

2. Missing ablation study. Because the dual-stage LoRA fine-tuning is a core contribution, more visual and numeric comparisons should be provided to demonstrate that such a design leads to prior performance. I see one ablation in the supplementary, but hope to see with/ and w/o each stage.


Minor suggestions:
1. Please add "Ours" to the proposed method in the figures when comparing with other methods (e.g., Figure 1).

2. Please keep the method's name and the position of zoomed regions in the video demo consistently. Please do not use brown color that is similar to the background, try some vivid ones like red, light yellow, or green for better visualization. Try to stop at images with the best visualization in the demo to show differences more clearly.

The paper can be better organized to make it more complete and convincing. I give my current rating based on the assumption that the dual-stage training mechanism brings performance improvements and is unique compared with existing techniques. I will change my rating according to authors' feedback.

---

> ### Author Rebuttal · Authors · 2025-07-30
>
> We sincerely thank this reviewer for the constructive comments. We hope our following point-to-point responses can address this reviewer's concerns. And we adopt VideoLQ4, as detailed in Appendix Section2, as our default test set for the reported experiments.
>
> **[W1]: Clarity of contributions**.
>
> **[Response]:** Thanks for the question, and we are sorry if we did not make our contributions clear enough in the main paper. Regarding the CSD loss [1], it draws on the idea of score distillation from pre-trained text-to-image diffusion models. It compares two predictions from the frozen base model: one conditioned on the text prompt and one unconditioned. The difference guides optimization toward outputs that better align with the textual semantics and exhibit higher visual quality. This loss is applied only in the enhancement stage, which focuses on detail enhancement; while in the consistency stage, we emphasize on temporal coherence, making CSD less relevant.
>
> In addition to the design of loss functions, the novelty of our proposed two-stage framework lies in the use of distinct training data, the Cross-Frame Retrieval (CFR) module, the integration of dual LoRA parameter spaces, and the alternative training strategy. Specifically, the consistency stage is trained on real-world videos with complex motion, enabling the use of consistency loss and focusing on temporal coherence. While the enhancement stage uses high-quality image data with synthetic motion. It is guided by inherited consistency loss components to retain learned priors, and also by the CSD loss to enhance fine-grained details. This decoupling allows us to define two specialized and trainable parameter spaces, each leveraging the strengths of different data types and loss functions, which is rarely explored in prior works. By alternating the two phases during iterative optimization, the framework works collaboratively to achieve both temporal consistency and high-quality detail enhancement.
>
> We acknowledge that the description of the CSD loss is not sufficiently detailed in the original manuscript, and we will add a clear explanation of its motivation and implementation in the revised version. We note that the contributions of dual stages can be reflected in the ablation study provided in our response to W2. We acknowledge that this key experiment is missing from the original manuscript, and we will include it in the revised version to improve clarity.
>
>
>
> **[W2]: Ablation of each stage**.
>
> **[Response]:**  We thank the reviewer for the insightful suggestion regarding an ablation study on the two stages. We conducted a dedicated experiment with all models trained from scratch, comparing (i) consistency-stage only, (ii) enhancement-stage only, and (iii) dual-stage design (ours). Due to the policy of NeurIPS rebuttal, we are unable to include visual results here. We will incorporate these results in the revised manuscript to provide a more comprehensive evaluation.
>
> |                         | MUSIQ ↑       | CLIP-IQA ↑     | MANIQA ↑       | Warping Error ↓       |
> |-------------------------|---------------|----------------|----------------|------------------------|
> | Consistency-stage only  | 50.5394       | 0.3052         | 0.2147         | **1.48 × 10⁻³**        |
> | Enhancement-stage only  | 65.8557       | 0.5420         | 0.3412         | 1.62 × 10⁻³            |
> | Dual-stage design (Ours)| **66.6174**   | **0.5475**     | **0.3791**     | 1.51 × 10⁻³            |
>
> These results confirm that the temporal consistency is primarily enhanced in the consistency stage, while fine spatial details are recovered in the enhancement stage; combining both yields the best overall performance. These findings substantiate the complementary contributions of each stage and justify the proposed dual‑stage design.
>
> **[W3] and [W4]: Suggestions of visual presentation**.
>
> **[Response]:**  We thank the reviewer for these valuable presentation suggestions. We will (i) label our method explicitly as “ours” in relevant figures, and (ii) adjust the demo video for better visual presentation.
>
> **[Q1]: The reason of choosing T2I model**.
>
> **[Response]:**  Thank you for the thoughtful question. We agree that recent video generation models (T2V) [2,3] have made remarkable progress and can offer stronger temporal consistency. However, we chose to train our model based on the T2I model primarily due to its superior image quality, which is important for VSR.
>
> To support our choice, we conduct a direct comparison by training our VSR model with a T2I model (SD2.1) and a T2V model (Wan2.1-1.3B). Both VSR models are trained on the same video dataset with identical training settings (*i.e.*, joint training with composite consistency and enhancement losses).  And the results are shown below:
>
> |         | MUSIQ ↑       | CLIP-IQA ↑     | MANIQA ↑       | Warping Error ↓       |
> |---------|---------------|----------------|----------------|------------------------|
> | Wan2.1  | 43.0117       | 0.3615         | 0.2641         | **1.47 × 10⁻³**        |
> | SD2.1   | **54.1642**   | **0.4873**     | **0.3428**     | 1.50 × 10⁻³            |
>
> We see that the T2I model (SD2.1) achieves much higher image quality while maintaining comparable temporal consistency (measured by warping error).  This result may stem from the following two factors:
>
> 1. **Training data**: The per-frame quality of video data used to train T2V models is generally lower than that of images used to train T2I models so that T2V models are harder to generate fine-grained visual details.
>
> 2. **Model architecture**: T2V models typically adopt video VAEs that compress both spatial and temporal dimensions. This leads to a heavier compression compared to the VAE used in T2I models. causing more information loss.
>
> While T2V models can offer better temporal coherence, VSR tasks require both high visual quality and temporal consistency. In our experiments, T2I-based models provide a better trade-off between the two goals. STAR [4] in Table 1 of the main paper is based on a T2V model (*i.e.,* CogVideo [5]), which also reflects this quality gap.
>
> **Reference:**
>
> [1] L. Sun, R. Wu, Z. Ma, S. Liu, Q. Yi, L. Zhang, "Pixel-level and Semantic-level Adjustable Super-resolution: A Dual-LoRA Approach," Proceedings of the IEEE/CVF Conference on Computer Vision and Pattern Recognition (CVPR), 2025.
>
> [2] W. Kong, Q. Tian, Z. Zhang, R. Min, Z. Dai, J. Zhou, J. Xiong, X. Li, B. Wu, J. Zhang, "HunyuanVideo: A Systematic Framework for Large Video Generative Models," arXiv preprint arXiv:2412.03603, 2024.
>
> [3] T. Wan, A. Wang, B. Ai, B. Wen, C. Mao, C.-W. Xie, D. Chen, et al., "Wan: Open and advanced large-scale video generative models," arXiv preprint arXiv:2503.20314, 2025.
>
> [4] R. Xie, Y. Liu, P. Zhou, C. Zhao, J. Zhou, K. Zhang, Z. Zhang, J. Yang, Z. Yang, Y. Tai, "STAR: Spatial-temporal augmentation with text-to-video models for real-world video super-resolution," Proceedings of the IEEE/CVF International Conference on Computer Vision (ICCV), 2025.
>
> [5] W. Hong, M. Ding, W. Zheng, X. Liu, J. Tang, "CogVideo: Large-scale pretraining for text-to-video generation via transformers," Proceedings of the International Conference on Learning Representations (ICLR), 2023.

---

> ### Author Response · Authors · 2025-08-04
>
> Dear Reviewer 1Rdb,
>
> Many thanks for your time in reviewing our paper and your constructive comments. We have submitted the point-to-point responses. We appreciate if you could let us know whether your concerns have been addressed, and we are happy to answer any further questions.
>
> Best regards,
>
> Authors of paper \#8760

---

> > ### Comment · Reviewer_1Rdb · 2025-08-05
> >
> > I appreciate the response from the authors, which has generally solved my questions and concerns. I will keep my positive feedback on this work.

---

> > > ### Author Response · Authors · 2025-08-05
> > >
> > > Many thanks for the support on our work! We will adopt your suggestions into the revision of the manuscript.
> > >
> > > Authors of paper \#8760

---

### Official Review · Reviewer_Bv1v · 2025-07-01

**Clarity:** 3
**Significance:** 3
**Originality:** 3
**Rating:** 4
**Confidence:** 4

**Summary:**

The authors propose a method called DLoRAL to achieve faster, more consistent, and detail-enhanced video super-resolution. DLoRAL consists of a CFR module, C-LoRA, and D-LoRA, which are designed to extract cross-frame complementary information, maintain temporal consistency, and restore image details, respectively. Experiments are conducted to demonstrate the effectiveness of DLoRAL. Compared to previous methods, DLoRAL not only achieves improved performance but also requires less inference time.

**Questions:**

See Weaknesses.

I may consider increasing the score once all concerns are addressed.

**Ethical Concerns:**

["NO or VERY MINOR ethics concerns only"]

**Final Justification:**

The author's rebuttal addresses my concerns, and given that the authors promise to incorporate the experimental results in the rebuttal into the revised version, my final score is borderline accept.

**Limitations:**

yes

**Quality:**

3

**Strengths And Weaknesses:**

Strengths:
1. The paper is well-written and easy to understand.
2. The design of Cross-Frame Retrieval is interesting, utilizing the top-k most similar K V pairs to compute cross-frame attention.
3. The proposed method employs a single-step diffusion model, resulting in faster inference speed compared to baselines.

Weaknesses:
1. Why use the image model SD for video super-resolution tasks? There are already several advanced video generation models, such as Hunyuan Video and Wan2.1 1.3B, with acceptable LoRA training costs. It seems more straightforward to perform video super-resolution tasks on video generation models.
    - Furthermore, based on my experience, the temporal inconsistency issue may stem from using SD for video generation. Using video generation models might significantly alleviate this issue and does not seem to require specific fine-tuning to enhance temporal generation capabilities, which is necessary when using SD. Therefore, I am concerned about the practical significance of the motivations behind this paper.
2. The proposed method is based on the UNet architecture, and I am uncertain if it can be adapted to the current DiT architecture.
3. How can it be ensured that the two designed LoRA implementations achieve true decoupling? The paper claims that D-LoRA is intended to learn high-frequency details, but since the supervision signal comes from video, it will inevitably learn motion information. Moreover, D-LoRA's loss also uses the loss from the Temporal Consistency Stage (C-LoRA).
4. Will training on constructed pseudo-video data for the enhancement stage affect performance? Since pseudo-videos lack reasonable motion information.
5. I noticed that the appendix includes some ablation studies on the training strategy; however, the paper lacks detailed ablations for each proposed module, as well as explanations of how these modules contribute to overall performance.
6. Does the proposed method maintain its faster inference speed when applied to multi-step diffusion models? Or, if the compared baseline uses a single-step diffusion model, is there a significant difference in inference speed compared to the proposed method?

---

> ### Author Rebuttal · Authors · 2025-07-30
>
> We sincerely thank this reviewer for the constructive comments. We hope our following point-to-point responses can address this reviewer's concerns. And we adopt VideoLQ4, as detailed in Appendix Section2, as our default test set for the reported experiments.
>
> **[[W1] and [Q1]]: The reason of choosing T2I model.**
>
> **[Response]:**  We agree that video generation models (T2V) [1,2] have made remarkable progress and can offer stronger temporal consistency. However, we employ the text-to-image (T2I) model primarily due to its superior image quality in the context of Video Super-Resolution (VSR).
>
> To support our choice, we conduct a direct comparison by training our VSR model using the SD 2.1 T2I model and the Wan2.1-1.3B T2V model on the same VSR training data with identical training settings (*i.e.*, joint training with composite consistency and enhancement losses). The results are shown below:
>
> |         | MUSIQ ↑       | CLIP-IQA ↑     | MANIQA ↑       | Warping Error ↓       |
> |---------|---------------|----------------|----------------|------------------------|
> | Wan2.1  | 43.0117       | 0.3615         | 0.2641         | **1.47 × 10⁻³**        |
> | SD2.1   | **54.1642**   | **0.4873**     | **0.3428**     | 1.50 × 10⁻³            |
>
> We see that SD2.1 achieves much higher image quality while maintaining comparable temporal consistency (measured by warping error).  This result may stem from the following two factors:
>
> 1. **Training data**: The per-frame quality of video data used to train T2V models is generally lower than that of images used to train T2I models so that T2V models are harder to generate fine-grained visual details.
>
> 2. **Model architecture**: T2V models typically adopt video VAEs that compress both spatial and temporal dimensions. This leads to a heavier compression compared to the VAE used in T2I models. causing more information loss.
>
> While T2V models can offer better temporal coherence, VSR tasks require both high visual quality and temporal consistency. In our experiments, T2I-based models provide a better trade-off between the two goals. STAR [3] in Table 1 of the main paper is based on a T2V model (*e.g.,* CogVideo [4]), which also reflects this quality gap.
>
> **[W2] and [Q2]: Adaption to DiT**.
>
> **[Response]:**  To answer this concern, we implement our approach on a DiT-based T2I model (*i.e.,* SD 3) and train the VSR model with and without our proposed strategies. The experimental results are shown below.
>
> |                       | MUSIQ ↑       | CLIP-IQA ↑     | MANIQA ↑       | Warping Error ↓       |
> |-----------------------|---------------|----------------|----------------|------------------------|
> | SD3 w/o our strategy  | 58.3651       | 0.4217         | 0.3243         | 1.62 × 10⁻³            |
> | SD3 w/ our strategy   | **69.0547**   | **0.5812**     | **0.4015**     | **1.55 × 10⁻³**        |
>
> We see that our method can be successfully integrated into the DiT architecture and consistently improves both perceptual quality and temporal consistency. This demonstrates the generality of our approach beyond UNet-based models (*i.e.,* SD 2.1).
>
> **[W3] and [Q3]: Decoupling of dual LoRA**.
>
> **[Response]:**  Thanks for the question. We would like to clarify that the term **decoupling** in our method refers to **functional specialization**, rather than entirely separate training processes.
>
> Please kindly note that D-LoRA is trained on high-quality images from the LSDIR dataset, not on real-world video data as used by C-LoRA. It enables D-LoRA to focus on enhancing perceptual details.  The temporal loss inherited from C-LoRA serves to retain the learned temporal priors and prevent forgetting, rather than to model motion.
>
> We further validate this design with two experiments. First, as shown in the table below, adding D-LoRA to a fixed C-LoRA during inference significantly improves visual quality (higher MUSIQ), while maintaining comparable temporal coherence (similar warping error). This demonstrates that D-LoRA effectively enhances perceptual quality without interfering with C-LoRA in maintaining temporal consistency.
>
> |                   | MUSIQ ↑       | CLIP-IQA ↑     | MANIQA ↑       | Warping Error ↓       |
> |-------------------|---------------|----------------|----------------|------------------------|
> | C-LoRA only       | 56.8002       | 0.3124         | 0.2296         | **1.49 × 10⁻³**        |
> | C+D LoRA (Ours)   | **66.6174**   | **0.5475**     | **0.3791**     | 1.51 × 10⁻³        |
>
> Second, the table below shows that removing synthetic motion and the consistency loss ($L_{opt}$) from D-LoRA training results in worse temporal consistency (*i.e.*, higher warping error), even though the visual quality (*e.g.*, MUSIQ) remains comparable.
>
> |                       | LPIPS ↓      | MUSIQ ↑       | CLIP-IQA ↑     | Warping Error ↓       |
> |-----------------------|--------------|----------------|----------------|------------------------|
> | W/o $L_{opt}$         | 66.5229      | 0.5243         | 0.3617         | 1.65 × 10⁻³            |
> | W/ $L_{opt}$ (Ours)   | **66.6174**  | **0.5475**     | **0.3791**     | **1.51 × 10⁻³**        |
>
> Therefore, though C-LoRA and D-LoRA share some common training elements, their functional roles are decoupled, where C-LoRA maintains the temporal consistency and D-LoRA enhances the details.
>
> **[W4] and [Q4]: Pseudo-video training data for enhancement stage**.
>
> **[Response]:**  While existing video datasets [5, 6, 7, 8] offer diverse motion patterns, they often suffer from limited visual quality compared to image datasets.  To prioritize perceptual quality in the enhancement stage, we use high-quality image data with synthetic motion as the training data. This design allows D-LoRA to focus on visual detail enhancement without being hindered by the lower quality of real video frames.
>
> However, this does not mean we ignore realistic motion. The consistency stage is trained on real videos with complex, natural motions, ensuring strong temporal priors. This decoupled design allows us to leverage the strengths of both data sources — real motion from videos and fine details from images — achieving a better overall performance of temporal coherence and perceptual quality.
>
> To assess the impact of this choice, we retrain the enhancement stage on the Pexel video dataset [5], one of the highest-quality public video datasets. As shown in the table below, training on real video datasets leads to reduced image quality (*e.g.,* lower MUSIQ) while maintaining comparable temporal consistency (*i.e.*, similar warping error). This confirms that pseudo-video training offers better perceptual quality without sacrificing consistency, and is thus more suitable for the enhancement stage.
>
> |                                 | MUSIQ ↑       | CLIP-IQA ↑     | MANIQA ↑       | Warping Error ↓       |
> |---------------------------------|---------------|----------------|----------------|------------------------|
> | Video-supervised enhancement stage | 57.5231       | 0.3658         | 0.2745         | **1.50 × 10⁻³**        |
> | Image-supervised enhancement stage | **66.6174**   | **0.5475**     | **0.3791**     | 1.51 × 10⁻³        |
>
> **[W5] and [Q5]: Ablation of each module**.
>
> **[Response]:**  Thank for the nice suggestion. In response, we add a new ablation study to analyze the contribution of each of our proposed modules:  (i) C-LoRA,  (ii) D-LoRA, and  (iii) CFR.  We compare the full model (denoted as ours) with three variants, each removing one module while keeping the others unchanged.
>
> |              | MUSIQ ↑       | CLIP-IQA ↑     | MANIQA ↑       | Warping Error ↓       |
> |--------------|---------------|----------------|----------------|------------------------|
> | Ours (Full)  | **66.6174**   | 0.5475         | **0.3791**     | 1.51 × 10⁻³            |
> | W/o CFR      | 64.5732       | 0.5148         | 0.3386         | 1.58 × 10⁻³            |
> | W/o C-LoRA   | 64.2623       | **0.5492**     | 0.3520         | 1.61 × 10⁻³            |
> | W/o D-LoRA   | 54.0769       | 0.3654         | 0.2471         | **1.48 × 10⁻³**        |
>
> As shown in the table above, removing either CFR or C-LoRA leads to weaker temporal consistency (*i.e.,* higher warping error), indicating their complementary roles in maintaining temporal coherence. In contrast, removing D-LoRA significantly impairs all perceptual metrics, confirming its core contribution to fine-grained detail enhancement.
>
> **[W6] and [Q6]: Adaption to multi-step diffusion models**.
>
> **[Response]:**  We thank the reviewer for the question. If we understand correctly, the concern is whether our faster inference speed mainly comes from using a single-step diffusion model. This is correct — our method is built on a one-step framework, which naturally runs faster than its multi-step counterparts. While this design provides clear efficiency benefits, our main contribution lies in enabling detail-rich and temporally consistent video super-resolution within this lightweight framework.
>
> To ensure a fair comparison, we also evaluate against other single-step baselines. As shown in Tables 1 and 2 of the main paper, our method achieves comparable inference speed to OSEDiff, a state-of-the-art single-step model, while delivering better perception quality and temporal consistency.
>
> **Reference:**
>
> [1] Kong et al. HunyuanVideo: Large Video Generative Models. arXiv, 2024.
>
> [2] Wan et al. WAN: Large-scale Video Generation. arXiv, 2025.
>
> [3] Xie et al. STAR: Text-to-Video Models for VSR. ICCV, 2025.
>
> [4] Hong et al. CogVideo: T2V Generation via Transformers. ICLR, 2023.
>
> [5] Pexels. https://www.pexels.com/
>
> [6] Nan et al. OpenVid-1M: High-Quality T2V Dataset. ICLR, 2025.
>
> [7] Bain et al. Frozen in Time: Joint Video-Image Encoder. ICCV, 2021.
>
> [8] Zhou et al. Upscale-A-Video: Diffusion for Real-World VSR. CVPR, 2024.

---

> ### Author Response · Authors · 2025-08-04
>
> Dear Reviewer Bv1v,
>
> Many thanks for your time in reviewing our paper and your constructive comments. We have submitted the point-to-point responses. We appreciate if you could let us know whether your concerns have been addressed, and we are happy to answer any further questions.
>
> Best regards,
>
> Authors of paper \#8760

---

> ### Comment · Reviewer_Bv1v · 2025-08-07
> **Reply to the rebuttal**
>
> Thank the authors for the rebuttal and additional experiments, which have addressed my concerns. If the authors commit to adding all the experimental results in the rebuttal to the revised version, I will raise the score to borderline accept.

---

> > ### Author Response · Authors · 2025-08-07
> >
> > We sincerely thank this reviewer for the positive feedback! We will definitely add the experimental results provided in the rebuttal to the revised version. You and the other reviewers’ constructive comments indeed helped us to shape this work much stronger.
> >
> > Authors of paper \#8760

---

### Official Review · Reviewer_mZuh · 2025-07-02

**Clarity:** 3
**Significance:** 3
**Originality:** 3
**Rating:** 4
**Confidence:** 4

**Summary:**

This paper addresses the challenge of Real-World Video Super-Resolution (Real-VSR), where the goals of generating rich spatial details and maintaining temporal consistency are often in conflict. The authors propose a novel framework, Dual LoRA Learning (DLORAL), which leverages a pre-trained Stable Diffusion model to perform high-quality VSR in a single diffusion step. The core idea is to decouple the learning of temporal consistency and spatial detail into two specialized, low-rank adaptation (LoRA) modules: a Consistency-LoRA (C-LORA) and a Detail-LoRA (D-LORA). The model first learns robust temporal priors from degraded inputs using a Cross-Frame Retrieval (CFR) module and the C-LORA. Subsequently, it enhances spatial details using the D-LORA while keeping the consistency modules frozen. These two training stages are alternated iteratively to find an optimal balance. During inference, the LoRAs are merged, enabling efficient, one-step restoration that, as experiments show, outperforms existing methods in both perceptual quality and speed.

**Questions:**

I believe the paper is strong, but my final evaluation could be improved if the authors address the following points:

- Regarding training complexity, could you please provide an ablation study or at least a discussion on the necessity of the alternating, dual-dataset training scheme? Specifically, how does this approach compare to a simpler baseline that performs joint training on a combined dataset, using a single, fixed loss function (e.g., $L_{enh}$ from Eq. 5) from the start? This would help clarify whether the performance gains are truly from the decoupled optimization or from other factors.

- The CFR module's reliance on only the single preceding frame (t−1) seems potentially fragile. How does the model perform in video scenes with significant challenges to this assumption, such as rapid motion, severe occlusions, or scene cuts where the t−1 frame provides little useful information? Have you considered extending the CFR module's temporal window to incorporate information from more distant frames (e.g., t−2, t−5)?

- The generation of motion via random pixel shifts on still images is an interesting choice. How crucial is this simulated motion for the performance of the Detail-LoRA? What happens if D-LORA is trained on high-quality still images alone, without the optical flow regularization term ($L_{opt}$)? Does this significantly degrade temporal consistency, or does the frozen C-LORA and CFR module provide sufficient temporal guidance on their own? Answering this could potentially simplify the data preparation pipeline significantly.

**Ethical Concerns:**

["NO or VERY MINOR ethics concerns only"]

**Final Justification:**

The added comparisons convincingly show that (i) the dual-stage, dual-dataset schedule outperforms a joint-training baseline, (ii) a temporal window = 1 in the CFR module offers the best trade-off between perceptual quality and temporal coherence without excessive memory/time cost, and (iii) the lightweight synthetic-motion curriculum is crucial for preserving temporal consistency during detail enhancement. These results directly address my earlier concerns about training complexity, window size, and data realism.

Some open questions remain—e.g., performance on extremely rapid-motion scenes and practical guidance for hyper-parameter tuning—but the paper now provides solid evidence of a state-of-the-art VSR method that combines high visual quality with single-step inference speed. I therefore raise my score to 4.

**Limitations:**

Yes, the authors have included a limitations section.

**Paper Formatting Concerns:**

None.

**Quality:**

3

**Strengths And Weaknesses:**

Strengths:

- The central contribution—decoupling the optimization of temporal consistency and spatial detail via a dual-LoRA, alternating training paradigm—is highly original and significant. This method directly confronts the fundamental trade-off that plagues many generative VSR models. The resulting ability to produce both temporally stable and detail-rich videos marks a notable advancement in the field.

- The paper demonstrates state-of-the-art performance across a wide range of quantitative metrics and on multiple challenging real-world and synthetic datasets. The qualitative results presented in Figures 3 and 4 are particularly impressive, showing clear superiority over competing methods in restoring facial structures and fine textures. Critically, the model achieves this while being remarkably efficient, performing inference in a single step, which is approximately 10x faster than other diffusion-based VSR models. This combination of quality and speed is a major strength.

- The technical execution is solid. The Cross-Frame Retrieval (CFR) module is well-designed, employing mechanisms like top-k attention and a learnable gating threshold to robustly aggregate information from noisy adjacent frames, rather than using a naive fusion. The dynamic, iterative training process with a smooth loss transition strategy is also a thoughtful detail to ensure stable convergence.

Weaknesses:

- The proposed training pipeline is notably complex, involving two separate stages, two different training datasets, alternating optimization, and a carefully scheduled loss interpolation for transitions. While this yields impressive results, the paper lacks an ablation study to justify this complexity. It is unclear how sensitive the final performance is to this specific training curriculum. A simpler joint-training approach with a well-weighted composite loss might potentially achieve comparable results with greater ease of implementation and training stability.

- The Cross-Frame Retrieval (CFR) module, which is critical for establishing consistency, operates with a sliding window of only two frames (the current and the immediately preceding one). This architectural choice may limit the model's ability to handle more complex temporal dynamics, such as occlusions, dis-occlusions, rapid non-linear motion, or re-emerging objects after several frames. The reliance on only the previous frame for temporal priors is a potential bottleneck for long-term consistency.

- For the detail enhancement stage, the authors generate "video sequences" by applying random pixel-level translations to single high-quality images from the LSDIR dataset. While this enables the use of an optical flow loss, this synthetic motion is highly simplistic and not representative of real-world camera or object motion. It is unclear if training D-LORA on this artificial data is truly beneficial or if it might bias the model in subtle ways.

---

> ### Author Rebuttal · Authors · 2025-07-29
>
> We sincerely thank this reviewer for the constructive comments and suggestion. We hope our following point-to-point responses can address this reviewer's concerns. And we adopt VideoLQ4, as detailed in Appendix Section2, as our default test set for the reported experiments.
>
> **[W1] and [Q1]: Clarification about the training complexity.**
>
> **[Response]:**  We appreciate the reviewer's comment. While our method involves two training stages and two datasets, we would like to clarify that each stage is purposefully paired with a single task-specific dataset and the dedicated losses, while the two stages are trained independently in an alternating manner. This design is actually not that complex, but rather a deliberate choice to decouple the optimization of temporal consistency and spatial detail via a dual-LoRA framework.
>
> Specifically, the C-LoRA is trained exclusively on degraded real-world videos to learn robust temporal priors. The D-LoRA is trained only on high-quality images (with synthetic motion) to enhance spatial detail.
> This decoupled training results in a stable training process and improved final performance than the joint-training approach with a mixed dataset and a combined loss, as shown in our experiments in Appendix Section2: Ablation studies on training strategies in DLoRAL. Observed from the Figure 1 of the Appendix, while a joint-training scheme with a carefully balanced loss may appear simpler, it will result in conflicts between competing objectives, leading to slower convergence and weaker overall performance. In contrast, our dual-space, dual-stage design separates the learning of detail and consistency, enabling higher visual quality and more stable temporal alignment.
>
> **[W2] and [Q2]: Concerns on CFR.**
>
> **[Response]:**  We sincerely thank the reviewer for raising this important concern. For challenging scenes where the $t{-}1$ frame provides limited useful information, the absolute attention mechanism in our CFR module selectively integrates relevant features while effectively filtering out noise from the previous frame. This design ensures that the reconstruction of the current frame remains robust without being adversely affected by irrelevant information.
>
> Regarding the temporal window size, we evaluate the use of a larger temporal window in the CFR module. And the corresponding results are shown below.
>
> |                          | MUSIQ ↑     | CLIP-IQA ↑     | MANIQA ↑       | Warping Error ↓       |
> |--------------------------|-------------|----------------|----------------|------------------------|
> | Temporal window = 0 (w/o CFR) | 64.5732     | 0.5148         | 0.3386         | 1.58 × 10⁻³            |
> | Temporal window = 1 (Ours)    | **66.6174** | **0.5475**     | **0.3791**     | 1.51 × 10⁻³        |
> | Temporal window = 3           | 65.4470     | 0.5452         | 0.3645         | 1.51 × 10⁻³        |
> | Temporal window = 5           | 62.1562     | 0.5019         | 0.3204         | **1.50 × 10⁻³**            |
>
> From the table, we observe that a temporal window size of 1 achieves a good balance between temporal consistency and visual quality. When the temporal window is increased to 5, there is a slight drop in the warping error, indicating better temporal consistency. However, this comes at the cost of noticeably reduced visual quality, as reflected by the decrease in metrics such as MUSIQ and CLIP-IQA. On the other hand, when the temporal window is set to 0 (*i.e.,* without CFR), the warping error is significantly worse, highlighting the importance of cross-frame interactions introduced by the CFR module.
>
> Furthermore, larger temporal window corresponds to higher computational cost. This is due to the increased number of cross-frame interactions (*i.e.,* cross-attention operation), which leads to quadratic growth in memory and computation requirements.
>
> In conclusion, while a larger temporal window can slightly improve temporal consistency, the trade-offs in visual quality and computational efficiency make it less attractive. Therefore, we adopt a temporal window of 1 in our final design.
>
> **[W3] and [Q3]: Training data of enhancement stage.**
>
> **[Response]:**  We appreciate the reviewer's thoughtful attention to the design of our training data for detail enhancement. Simulated motion is used to prevent the model from forgetting previously learned temporal consistency. To further validate its effectiveness, we retrain the model by adopting high-quality still images and removing the optical flow loss in the second stage. The results are shown in the table below.
>
> |                         | MUSIQ ↑     | CLIP-IQA ↑     | MANIQA ↑       | Warping Error ↓       |
> |-------------------------|-------------|----------------|----------------|------------------------|
> | W/ simulated motion (Ours)   | **66.6174** | **0.5475**     | **0.3791**     | **1.51 × 10⁻³**        |
> | W/o simulated motion         | 66.5229     | 0.5243         | 0.3617         | 1.65 × 10⁻³            |
>
>
> The two models achieve comparable scores on visual metrics, indicating similar perceptual quality. However, the model trained without simulated motion exhibits a noticeably higher warping error, suggesting degraded temporal consistency. This gap confirms that—even though the synthetic motion is simplistic—it plays an important role in preventing the model from forgetting previously learned temporal consistency during detail enhancement.

---

> ### Author Response · Authors · 2025-08-04
>
> Dear Reviewer mZuh,
>
> Many thanks for your time in reviewing our paper and your constructive comments. We have submitted the point-to-point responses. We appreciate if you could let us know whether your concerns have been addressed, and we are happy to answer any further questions.
>
> Best regards,
>
> Authors of paper \#8760

---

> ### Comment · Reviewer_mZuh · 2025-08-05
>
> Thank you for the careful rebuttal and new ablations. The added comparisons convincingly show that (i) the dual-stage, dual-dataset schedule outperforms a joint-training baseline, (ii) a temporal window = 1 in the CFR module offers the best trade-off between perceptual quality and temporal coherence without excessive memory/time cost, and (iii) the lightweight synthetic-motion curriculum is crucial for preserving temporal consistency during detail enhancement. These results directly address my earlier concerns about training complexity, window size, and data realism.
>
> Some open questions remain—e.g., performance on extremely rapid-motion scenes and practical guidance for hyper-parameter tuning—but the paper now provides solid evidence of a state-of-the-art VSR method that combines high visual quality with single-step inference speed. I therefore raise my score to borderline accept.

---

> > ### Author Response · Authors · 2025-08-06
> >
> > We sincerely thank this reviewer for the positive and comprehensive comments on work! Your constructive suggestions are truly helpful for us to improve this work, and we will surely adopt them into the revision of the manuscript.
> >
> > Best regards,
> >
> > Authors of paper \#8760

---

### Official Review · Reviewer_dmAC · 2025-07-02

**Clarity:** 3
**Significance:** 3
**Originality:** 3
**Rating:** 4
**Confidence:** 4

**Summary:**

This paper is interesting. The authors add two LORA to enhance the temporal consistency and details of the original one step diffusion SR network.

**Questions:**

Perhaps ablation studies can be given to show the contribution of each LORA.

**Ethical Concerns:**

["NO or VERY MINOR ethics concerns only"]

**Limitations:**

Can such design be applied to other image-based diffusion networks? such as image generation network.

**Quality:**

3

**Strengths And Weaknesses:**

Experimental results show obvious improvments with slight computational increments.

Strengths
1. Decoupled Learning for Conflicting Objectives: By separating temporal consistency (C-LoRA) and spatial detail (D-LoRA) into dedicated modules, DLoRAL avoids the trade-off inherent in joint optimization, allowing each component to specialize in its target.
2. Effective Extraction of Temporal Priors: The CFR module leverages inherent consistency in LQ videos (despite noise/blur) to aggregate structure-aligned cross-frame information, providing stable guidance for both modules.
3. Efficient One-Step Inference: Built on a one-step diffusion framework, DLoRAL reduces inference time by ~10× compared to multi-step diffusion methods (e.g., Upscale-A-Video, MGLD), with comparable or better quality.
4. Strong Performance Across Metrics: DLoRAL achieves state-of-the-art results in perceptual metrics (MUSIQ, CLIPIQA), fidelity (PSNR, SSIM), and temporal consistency (warping error), balancing detail and coherence effectively.
Alternate Training Strategy: Iterative optimization of consistency and detail stages, with smooth loss transitions, ensures each module converges without interfering with the other.

Weaknesses
1. Limitations of VAE Compression: Inherits the 8× downsampling VAE from Stable Diffusion, which struggles to restore ultra-fine details (e.g., small text) due to aggressive compression.
2. Temporal Coherence Disruption: The VAE’s heavy compression may disrupt temporal consistency in LQ frames, making it harder to extract robust temporal priors. A VAE specifically designed for Real-VSR could mitigate this.
3. Dependence on Pre-trained Models: Relies on the pre-trained Stable Diffusion backbone, which may limit generalization to highly specialized video degradation types not covered by the original model’s training data.

---

> ### Author Rebuttal · Authors · 2025-07-29
>
> We sincerely thank this reviewer for the constructive comments and suggestion. We hope our following point-to-point responses can address this reviewer's concerns. And we adopt VideoLQ4, as detailed in Appendix Section2, as our default test set for the reported experiments.
>
> **[W1 and W2]: Limitations of VAE.**
>
> **[Response]**:  We sincerely thank the reviewer for pointing out the limitation of using the pre-trained VAE from the Stable Diffusion (SD) model. This is indeed a well-known issue, which we have also explicitly discussed and identified as our limitation in our manuscript (Page 9, Lines 339–343). The aggressive 8× downsampling in the VAE can compromise the reconstruction of fine structures (*e.g.,* small text and textures). Additionally, a heavily compressed VAE may struggle to extract accurate temporal priors. Therefore, improving the VAE is an important research direction, which we plan to explore in future work.
>
> In this paper, however, we aim to strike a balance between leveraging the natural image prior of SD model and achieving temporal consistency for video super-resolution (VSR) task; thus, the original VAE is kept for compatibility with the pre-trained UNet. To jointly maintain visual quality and temporal consistency within this framework, we propose Cross-Frame Retrieval and Dual-LoRA Learning. As shown in Table 1 and Figure 4 of the main paper, our method achieves strong results across both consistency and quality metrics, demonstrating its ability to maintain temporal coherence while preserving high visual quality under the heavily compressed VAE.
>
> Looking forward, recent works, such as TVT [1], have attempted to employ a VAE with a lower downsampling rate in SD-based models. These approaches significantly mitigate the challenges of reconstructing fine structures and alleviating temporal coherence disruption caused by severe compression, while retaining the generative capability of pre-trained diffusion models. We find this direction highly promising and plan to explore VAE designs specifically tailored for Real-VSR, such as TVT-like strategies, in our future work. This line of research is complementary to our current work and may further enhance the applicability of diffusion-based VSR models in real-world scenarios.
>
> **[W3]: Generalization capability.**
>
> **[Response]:**  We sincerely thank the reviewer for raising this important concern about the generalization ability of our method. Our approach builds upon the pre-trained SD model to leverage its strong natural image generative priors, which have been validated to have high generative capacity in prior SR tasks [2,3,4].
>
> We fully acknowledge that the SD model was trained on large-scale image datasets, which may not cover many real-world video degradation types. This potential distribution mismatch could indeed affect generalization in unseen scenarios. To address this limitation and enhance the robustness of our model, instead of using SD’s original training data, we construct dedicated datasets for Real-VSR training.  These datasets simulate diverse degradation types and scene content, aiming to better align the Real-VSR needs. To further evaluate the generalization performance of our method, we conduct the following experiments:
>
> 1. **Cross-dataset evaluation**:
> We extensively test DLoRAL on datasets with varying degradation characteristics, including synthetic benchmarks (UDM10, SPMCS) and real-world datasets (RealVSR, VideoLQ). RealVSR contains mobile-captured videos, while VideoLQ features authentically degraded Internet content. As shown in Table 1 and Figure 3, our method consistently achieves strong performance across all datasets, demonstrating robustness under a wide range of degradation patterns.
>
> 2. **User study**:
> We also perform user studies (Figure 3 of the Appendix) to assess the perceptual quality in challenging conditions, such as low light, fast motion, and compression artifacts. The results indicate that DLoRAL maintains good perceptual quality and generalizes well in these difficult scenarios.
>
> Nonetheless, we agree that there can still be failure cases for degradations outside of our training data.
>
> **[Q1]: More ablation studies of each LoRA.**
>
> **[Response]:**  We thank the reviewer for the insightful suggestion regarding an ablation study on the two LoRA branches. We conduct a dedicated experiment with all models trained from scratch, comparing (i) C-LoRA only, (ii) D-LoRA only, and (iii) the full C+D configuration. The results are presented in the table below.
>
> |                  | MUSIQ ↑   | CLIP-IQA ↑       | MANIQA ↑         | Warping Error ↓          |
> |------------------|-----------|------------------|------------------|---------------------------|
> | C-LoRA only      | 50.5394   | 0.3052           | 0.2147           | **1.48 × 10⁻³**           |
> | D-LoRA only      | 65.8557   | 0.5420           | 0.3412           | 1.62 × 10⁻³              |
> | C+D LoRA (Ours)  | **66.6174** | **0.5475**      | **0.3791**       | 1.51 × 10⁻³              |
>
> The above results confirm that C-LoRA primarily enhances temporal consistency, while D-LoRA recovers fine spatial details; combining both yields the best overall performance. These findings substantiate the complementary contributions of each LoRA and justify the proposed dual‑stage design.
>
> **[Q2]: Application to other networks.**
>
> **[Response]:**  If we correctly understand the reviewer's question, it concerns whether our proposed design can be applied to other diffusion networks, such as DiT-based models (*e.g.*, Stable Diffusion 3). To verify this, we conduct additional experiments using Stable Diffusion 3 as the pre-trained model, one model trained with our proposed strategies (*i.e.*, CFR and Dual-LoRA learning) and the other model trained without the strategies.
>
> |                    | MUSIQ ↑     | CLIP-IQA ↑     | MANIQA ↑       | Warping Error ↓       |
> |--------------------|-------------|----------------|----------------|------------------------|
> | SD3 w/ our strategy  | **69.0547** | **0.5812**     | **0.4015**     | **1.55 × 10⁻³**        |
> | SD3 w/o our strategy | 58.3651     | 0.4217         | 0.3243         | 1.62 × 10⁻³            |
>
>
> The results show that our method can be successfully integrated into other diffusion models and leads to improved VSR results.
>
> **Reference:**
>
> [1] Q. Yi, S. Li, R. Wu, L. Sun, Y. Wu, L. Zhang, "Fine-structure Preserved Real-world Image Super-resolution via Transfer VAE Training," Proceedings of the IEEE/CVF International Conference on Computer Vision (ICCV), 2025.
>
> [2] J. Wang, Z. Yue, S. Zhou, K. C. Chan, C. C. Loy, "Exploiting Diffusion Prior for Real-world Image Super-resolution," International Journal of Computer Vision (IJCV), 2024.
>
> [3] X. Lin, J. He, Z. Chen, Z. Lyu, B. Dai, F. Yu, Y. Qiao, W. Ouyang, C. Dong, "DiffBIR: Toward Blind Image Restoration with Generative Diffusion Prior," Proceedings of the European Conference on Computer Vision (ECCV), 2024.
>
> [4] L. Sun, R. Wu, Z. Ma, S. Liu, Q. Yi, L. Zhang, "Pixel-level and Semantic-level Adjustable Super-resolution: A Dual-LoRA Approach," Proceedings of the IEEE/CVF Conference on Computer Vision and Pattern Recognition (CVPR), 2025.

---

> ### Author Response · Authors · 2025-08-04
>
> Dear Reviewer dmAC,
>
> Many thanks for your time in reviewing our paper and your constructive comments. We have submitted the point-to-point responses. We appreciate if you could let us know whether your concerns have been addressed, and we are happy to answer any further questions.
>
> Best regards,
>
> Authors of paper \#8760

---

### Comment · Area_Chair_pc2z · 2025-08-05
**Kind remind for author-reviewer discussion**

Dear Authors and Reviewers,

Thank you for submitting and reviewing the papers to contribute to the conference. This is a kind remind that the due date of author-reviewer discussion is coming soon. Please participate the discussion to clarify paper statement or concerns.

Thanks!

AC

---

> ### Author Response · Authors · 2025-08-05
>
> Dear AC,
>
> Many thanks for your kind reminder. We are eager to hear from the reviewers for their feedback and further questions to our responses.
>
> Best regards,
>
> Authors of paper \#8760

---

### Note · Authors · 2025-08-13

Dear Reviewers and Area Chairs,

We sincerely thank all reviewers for their valuable feedback and engagement during the rebuttal phase. We are encouraged by the reviewers' positive comments and recognition on our paper's novelty (`dmAC`, `mZuh`, `Bv1v`), effectiveness (`dmAC`, `mZuh`, `Bv1v`), and presentation (`Bv1v`, `1Rdb`). We will also adopt all the reviewers’ constructive comments and suggestions in the revision of our manuscript.

In summary, we propose a Dual LoRA Learning (DLoRAL) paradigm to train an effective SD-based one-step diffusion model for video super-resolution. Our approach can not only generate rich spatial details but also ensure temporal consistency, which is a long-standing challenge in video super-resolution. We hope our work can be helpful for peers in the community to develop more effective and efficient methods in this field.

Best regards,

Authors of Paper \#8760

---

### Decision · Program_Chairs · 2025-09-17

**Decision:**

Accept (poster)

**Comment:**

This paper tackles Real-World Video Super-Resolution (Real-VSR), where generating fine spatial details often conflicts with maintaining temporal consistency. The authors introduce Dual LoRA Learning (DLORAL), a framework built on a pre-trained Stable Diffusion model that achieves high-quality VSR in a single diffusion step. DLORAL separates the learning of temporal consistency and spatial detail into two LoRA modules: C-LoRA for consistency and D-LoRA for detail. Using a Cross-Frame Retrieval module, C-LoRA learns temporal priors, after which D-LoRA enhances spatial details while freezing consistency learning. The two modules are trained alternately to strike a balance, then merged at inference for efficient one-step restoration. Experiments show DLORAL surpasses existing methods in both perceptual quality and speed.

Reviewers' primary concerns revolved around the limitations of VAE compression, the complexity of the training pipeline, and the explanation of the loss term. The author provided a careful rebuttal and new ablations, which successfully addressed most concerns.

Overall, the paper is recommended for acceptance. Some questions still need to be addressed, as the reviewer mZuh mentioned.